# Anatomically Constrained ResNets Exhibit Opponent Receptive Fields; So What?

**Ethan Harris**∗† **Daniela Mihai**∗† **Jonathon Hare**∗†

## Abstract

Primate visual systems are well known to exhibit varying degrees of bottlenecks in the early visual pathway. Recent works have shown that the presence of a bottleneck between 'retinal' and 'ventral' parts of artificial models of visual systems, simulating the optic nerve, can cause the emergence of cellular properties that have been observed in primates: namely centre-surround organisation and opponency. To date, however, state-of-the-art convolutional network architectures for classification problems have not incorporated such an early bottleneck. In this paper, we ask what happens if such a bottleneck is added to a ResNet-50 model trained to classify the ImageNet data set. Our experiments show that some of the emergent properties observed in simpler models still appear in these considerably deeper and more complex models, however, there are some notable differences particularly with regard to spectral opponency. The introduction of the bottleneck is experimentally shown to introduce a small but consistent shape bias into the network. Tight bottlenecks are also shown to only have a very slight affect on the top-1 accuracy of the models when trained and tested on ImageNet.

## 1 Introduction and Motivation

Convolutional Neural Networks (CNNs) with a bottleneck designed to mimic the anatomical constraint imposed by the optic nerve have recently been shown to learn representations which more closely align with those found in primate visual systems. Specifically, Lindsey et al. [2019] showed that a reduction in the number of channels or convolutional neurons in the second layer of a CNN gives rise to centre-surround receptive fields in the first two layers followed by orientation selectivity in subsequent layers. Lindsey et al.'s experiments were based on a relatively simple model of the visual system (aka 'Retina-Net') constructed with a model of the retina (the first two layers up to the bottleneck), a model of the ventral stream (stacked convolutional layers) and a pair of fully connected layers to produce a classification. These models were trained on a greyscale version of the CIFAR-10 data set [Krizhevsky et al., 2009]. Harris et al. [2019, 2020] subsequently showed that the same network architecture (albeit with colour input) learns a simple opponent colour code in the bottleneck layer followed by a reduction in colour opponency in later layers. Harris et al. [2020] further demonstrated that the general trend that the bottleneck induces opponency was true across a range of data sets from CIFAR-10 to ImageNet [Russakovsky et al., 2015].

Modern neural networks for solving large-scale classification problems like ImageNet look very different to the aforementioned 'Retina-Net' and its colour variant, and do not incorporate any form of bottleneck near the input layer. At the same time, other recent research has demonstrated that there may be benefits to better modelling of the visual system in state-of-the-art models. For example, Geirhos et al. [2019] show that CNNs trained on ImageNet are biased towards texture information in contrast to Human observers who make classifications predominantly according to shape information.

---

∗Authors contributed equally

†Vision, Learning and Control Group, Electronics and Computer Science, University of Southampton, {ewah1g13, adm1g15, jsh2}@ecs.soton.ac.uk

2nd Workshop on Shared Visual Representations in Human and Machine Intelligence (SVRHM), NeurIPS 2020.

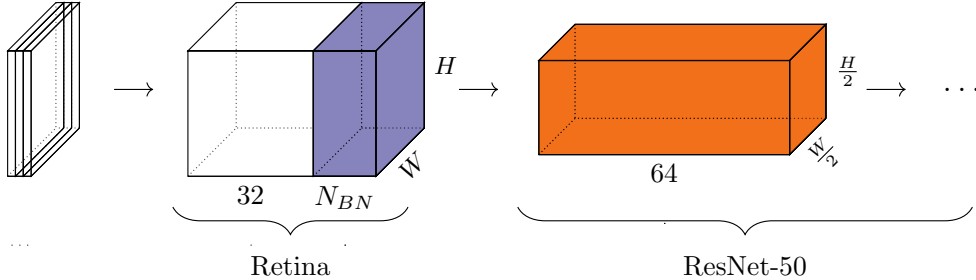

Figure 1: Anatomically constrained ResNet-50 model diagram. The only modification to the ResNet-50 architecture is that it expects an input with $N_{BN}$ channels. We trained such models on the ImageNet data set for $N_{BN} \in \{1, 2, 4, 8, 16, 32\}$.

They subsequently show that promoting a shape bias in trained models can improve both accuracy and robustness. Dapello et al. [2020] show that simulating the function performed by the primary visual cortex over the input increases robustness to adversarial attacks in which images which have been imperceptibly (to a human) adjusted such that the classification given by the trained network is no longer correct. Their model for simulating V1 (V1Block) is based on the classical observations (for example by Hubel and Wiesel [2004]) that the receptive fields of cells in V1 are orientation selective (modelled as Gabor filters) and that these cells can be described as either simple or complex (modelled with different acitvation functions). Parameters for the Gabor filters in the V1Block are sampled according to known distributions from the neuroscience literature rather than being learned. The model also assumes that the input to the V1Block is just the normalised pixel values (subtract 0.5 and divide by 0.5 for each RGB colour channel) and eschews any notion of processing performed by the retina itself, or by the Lateral Geniculate Nucleus (LGN).

Since convolutional neurons already reflect many of the functional properties of cells found in the early visual system, it is prudent to ascertain whether more simple modifications, such as the bottleneck explored by Lindsey et al. [2019] and Harris et al. [2019, 2020], can be used to the same effect. This paper starts to ask what happens if we place a bottleneck at the beginning of a performant classification network trained on the ImageNet data set. We first show that the bottleneck induces similar emergence of relevant cell types, but with some important differences. In particular, we find that very tight bottlenecks give rise to cells that are perhaps better described as luminance opponent rather than colour opponent. In addition, we find that the stark differences between receptive fields of cells in networks with different bottlenecks, observed by Lindsey et al. [2019], are more nuanced with centre-surround receptive fields seeming to emerge for all bottlenecks. We further show that the introduction of the bottleneck introduces a small but consistent shape bias. Finally, we consider the adversarial robustness of our models, showing that the bottleneck slightly improves robustness to natural adversarial examples but actually reduces robustness measured as a worst-case accuracy following multiple attacks.

## 2   Early Visual Representations in Anatomically Constrained ResNets

In order to add a bottleneck to an existing model, we propose pre-pending the 'Retina' portion of the model from Lindsey et al. [2019]. This consists of two convolutional layers with 32 channels and $N_{BN}$ channels respectively, where $N_{BN}$ is the bottleneck width. Unlike the model of Lindsey et al. [2019] which uses a kernel size of 9, we choose the kernel size to ensure a large enough receptive field whilst preserving consistency with the first layer of the target model where possible (e.g. 7 for a ResNet). Padding is chosen such that the resolution of the output is the same as the resolution of the input. In this way the bottleneck acts as a kind of parameterised pre-processing of the input. We trained constrained variants of a ResNet-50 (depicted in Figure 1) for the range of bottleneck widths used by Lindsey et al. [2019] ($N_{BN} \in \{1, 2, 4, 8, 16, 32\}$) on the ImageNet Large Scale Visual Recognition Challenge (ILSVRC) [Russakovsky et al., 2015] data set. We trained three repeats of each model variant, giving 18 models in total. The training regime is described in Appendix D.

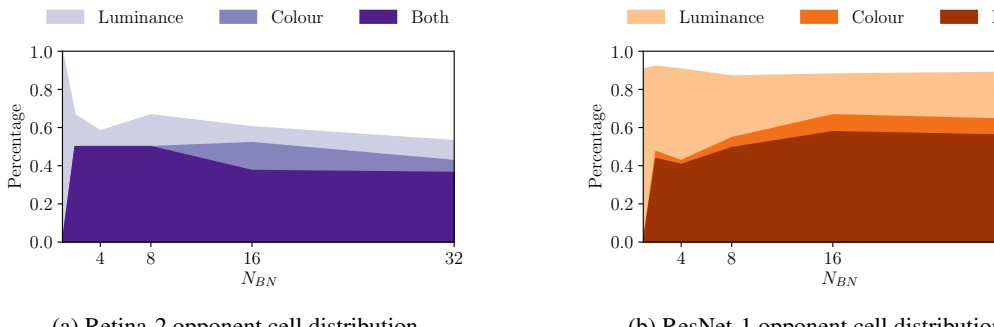

(a) Retina-2 opponent cell distribution        (b) ResNet-1 opponent cell distribution

Figure 2: Colour and luminance opponency distributions in the Retina-2 and ResNet-1 layers of anatomically constrained ResNet-50 models.

We first computed the distribution of opponent cell types following the approach from Harris et al. [2019, 2020] in the second retina layer (Retina-2) and first ResNet layer (ResNet-1). This approach consists of presenting a set of stimuli which vary in hue to the cell in order to obtain a hue response curve. The cell is colour opponent if the response curve crosses the baseline response of the cell to a zero stimulus. Interestingly, our results showed that networks with $N_{BN} = 1$ have no colour opponent cells in either of the layers we studied. This contrasts with the finding of Harris et al. [2019, 2020], who showed that almost all cells in the retina layers of their more constrained networks were colour opponent. We presume that the range of colour stimuli in ImageNet is sufficiently broad that when restricted to one channel, better performance can be obtained with a luminance opponent encoding than a colour opponent one. To test this, we additionally present a set of stimuli which vary in luminance (i.e. full field grey-scale stimuli with values ranging from zero to one) and perform the same process of comparing to the baseline (response of the cell to constant stimulus of $0.5$ for this experiment) to infer luminance opponency. Note that any cell whose response is linear in the sense that extremes of input value correspond to extremes of response will be classified as luminance opponent. It is therefore appropriate to say that any cells which are not luminance opponent are to some degree non-linear.

Figures 2a and 2b give the results for this experiment in the Retina-2 and ResNet-1 layers respectively. The results show that the vast majority of colour opponent cells are also luminance opponent, particularly in the more constrained networks. In the Retina-2 layer, there is an increase in cells which are luminance opponent, but not colour opponent, for networks with $N_{BN} < 4$. This in turn suggests an increase in the linearity of cell response for these networks, which accords with the findings of Lindsey et al. [2019], Harris et al. [2020]. Almost all cells in the first ResNet layer are luminance opponent. In both layers, approximately half of the cells are also colour opponent in networks with all but the tightest bottlenecks, where the percentage of colour opponent cells drops to zero. Ultimately, the pattern in the colour opponent cell distribution shown by Harris et al. [2019, 2020] is replaced with a similar pattern in luminance opponent cell distribution in these networks. We explore these cells further and find some evidence that they exhibit centre-surround organisation in Appendix A.

In the neuroscience literature, the distinction between luminance opponency and colour opponency is often unclear. For example, it has been argued that many of the cells which preferred achromatic stimuli in Monkey V1, described by Hubel and Wiesel [1968], may be better described as colour opponent but with imbalanced cone response such that they are also responsive to luminance [Johnson et al., 2001, Lennie et al., 1990, Schluppeck and Engel, 2002, Shapley and Hawken, 2011]. We therefore must ascertain whether the luminance opponent cells are still able to convey valuable information regarding the colour of the input. To do this, we propose performing gradient ascent on a random input in Fourier space, following Olah et al. [2017], to obtain super stimuli which the networks classify with high confidence as a range of classes which are characterised by a particular colour. Figure 3 shows these stimuli for a range of fruit classes in the ImageNet data set for models with $N_{BN} = 1$ and $N_{BN}$. From the figure we can see that the networks with a single ($N_{BN} = 1$) luminance opponent channel in the bottleneck layer still make use of colour to inform their decisions, learning the characteristic red of a strawberry, yellow of a lemon, green of an apple, and so on. This is possible since these luminance opponent cells may still have a broad non-opponent hue response that uniquely encodes colour over at least part of the hue wheel.

| $N_{BN}$ | Strawberry | Lemon | Apple | Orange | Pomegranate | Pineapple |

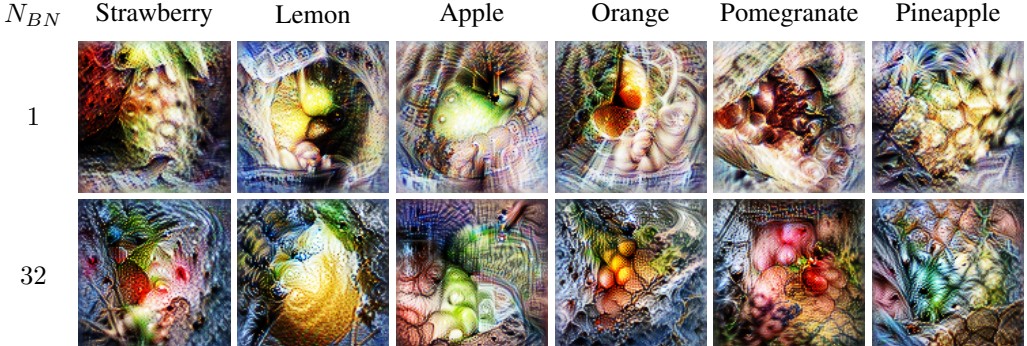

Figure 3: Super stimuli for various classes of fruit for networks with $N_{BN} = 1$ and $N_{BN} = 32$ obtained by gradient ascent in Fourier space following Olah et al. [2017]. Even when the networks are restricted to a single channel ($N_{BN} = 1$), they still learn a strong representation of colour.

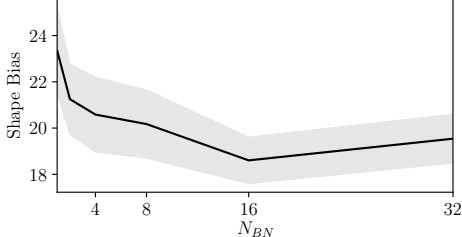

Figure 4: The shape bias (following Geirhos et al. [2019]) across bottleneck widths.

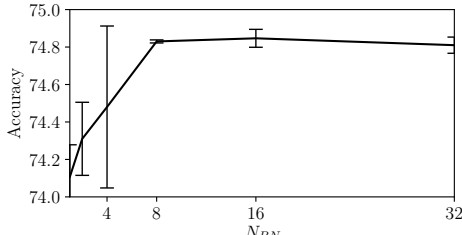

Figure 5: Top-1 accuracy on the ILSVRC validation set across different bottleneck widths.

It is clear from Figure 3 that neither of the models is particularly biased towards shape in that the super stimuli have appropriate textures and colours but not shapes. To quantify this, we now measure the shape bias of the models. Following Geirhos et al. [2019], we evaluate the models on the 'StylizedImageNet' data set [3], which contains images with a texture-shape cue conflict, requiring networks to recognise objects based on shape rather than texture. Figure 4 shows that networks with tighter bottlenecks exhibit a small increase in shape bias. The model with a bottleneck width of 1 trained on ImageNet achieves the highest shape bias, outrunning a standard (i.e. without our modification) ResNet-50 model. Note, however, that the shape bias of all of the models is far lower than the models trained with the approach from Geirhos et al. [2019].

## 3   What Effect Does a Bottleneck Have on Performance?

In this section we explore whether the changes in function induced by the bottleneck correspond to any tangible improvement in performance. The most obvious first step is to consider the effect of the bottleneck on the accuracy of the model. Figure 5 shows that networks with bottleneck width of 4 or greater all achieve validation set performance that is within the margin of error (one standard deviation) of the others and that networks with tighter bottlenecks show a small but clear drop in performance. Importantly, all of the networks perform within 2% of the pre-trained ResNet-50 without our modification provided in torchvision (76.15%), which has minor differences in the training procedure that might explain this. This contrasts with the networks considered by Dapello et al. [2020] which show a larger reduction of around 4.5% reaching an accuracy of 71.7 (although again note that there are differences in the training procedure). Note that in addition to small differences in training procedure, our bottleneck layers do not make use of modern conventions such as BatchNorm which are known to provide important performance benefits, particularly in deeper networks.

---

[3]Available from `https://github.com/rgeirhos/Stylized-ImageNet`

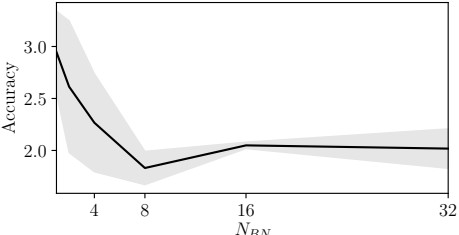

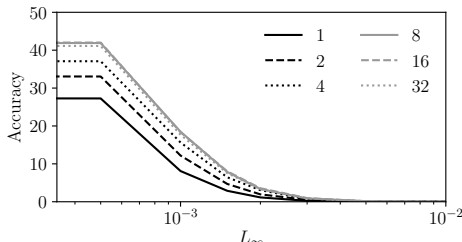

Figure 6: Robustness to natural adversarial examples from the ImageNet-A data set [Hendrycks et al., 2019] as a function of bottleneck width.

Figure 7: Robust accuracy of different bottlenecks (line styles) against $L_\infty$ adversarial attack budgets.

**Robustness to natural adversarial examples**   We now consider robustness to the ImageNet-A data set [Hendrycks et al., 2019], which contains 7500 naturally occurring images which are consistently mis-classified by pre-trained models. The images correspond to 200 ImageNet classes in which normal models (ResNet50, VGG16, etc) have a top-1 accuracy of more than 90% using the ImageNet validation data set. The ImageNet-A data set typically results in a top-1 accuracy of around 2-3% on the same models. Figure 6 shows the top-1 accuracy of our models on the ImageNet-A data set. Figure 11 from the appendix shows the Area Under the Response Rate Accuracy curve (AURRA) which measures how calibrated the models are (how a model's confidence in its prediction relates to its accuracy). Both of these figures illustrate that the tighter bottlenecks do induce slightly better performance on this task.

**Robustness to $L_\infty$ constrained artificial attacks**   To study a more general notion of adversarial robustness, we follow the guidelines given by Carlini et al. [2019]. We use FoolBox [Rauber et al., 2017, 2020] to compute the worst case performance of the models under a range of attacks. We explore a range of perturbations using different $L_\infty$ thresholds between 0 and 1.0. The attacks include Project Gradient Descent, which was utilised by Dapello et al. [2020] to demonstrate improved robustness of their model. A full description of the specific attacks is given in Appendix C. Figure 7 shows the robust accuracy, the worst-case performance following all attacks, of the models across different $L_\infty$ perturbation budgets. Results were computed by sampling 10000 images from the ImageNet validation set uniformly for each class. The results show that the increased opponency and small increase in shape bias actually correspond to a decrease in robustness to targeted adversarial examples. In Appendix B, we further show that the CIFAR-10 trained models from Harris et al. [2019, 2020] show no significant difference in adversarial robustness as a function of bottleneck.

## 4   Discussion

In this paper we have explored the addition of a bottleneck to ImageNet trained ResNet-50 models. We have shown that cells in the bottleneck layer of models with tight bottlenecks learn opponent receptive fields in accordance with the CIFAR-10 results of Harris et al. [2019, 2020]. In contrast to those results, the opponent cells we find here are best described as luminance opponent rather than colour opponent. That said, we have shown that these luminant opponent cells still exhibit some form of spectral response and that, even in the extreme case of $N_{BN} = 1$, the model is able to learn about colour. Whether the cells in networks with tight bottlenecks have centre-surround receptive fields of the kind observed by Lindsey et al. [2019] is unclear, although there is some evidence that they do. It is, however, clear that the Retina-2 cells respond more linearly in networks with tight bottlenecks, echoing the finding of Lindsey et al. [2019]. We have further shown that these differences in function correspond to a small but consistent increase in the shape bias of the networks.

Our investigation into the performance of the different bottlenecked models is surprising in the sense that it perhaps asks more questions than it answers. At the outset our hypothesis was that the bottleneck might encourage both shape bias as well as adversarial robustness; the typical nature of centre-surround opponent cells observed in prior studies would for example suggest that the early network would perform isotropic band-pass filtering (like a Laplacian of Gaussian) which one

presumes would naturally bias to towards shape representations[4]. In terms of accuracy, the different bottlenecks only have a mild effect with the tightest bottlenecks (below 8) having slightly worse performance. The results of testing the models on natural adversarial examples indicates that the tightest bottlenecks do perform better on the ImageNet-A data set (both in terms of accuracy and calibration). However, it is unclear whether this increase corresponds to an increase in robustness or merely indicates that the type of function learned differs for tighter bottlenecks. In any case, the worst case accuracy, computed after a set of targeted adversarial attacks, shows that adversarial robustness actually decreases slightly as the bottleneck is tightened.

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

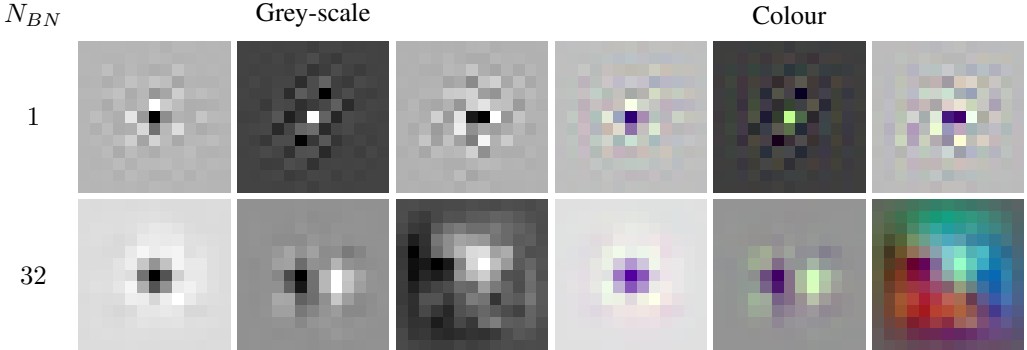

Figure 8: Grey-scale and colour receptive field visualisations for networks with $N_{BN} = 1$ and $N_{BN} = 32$.

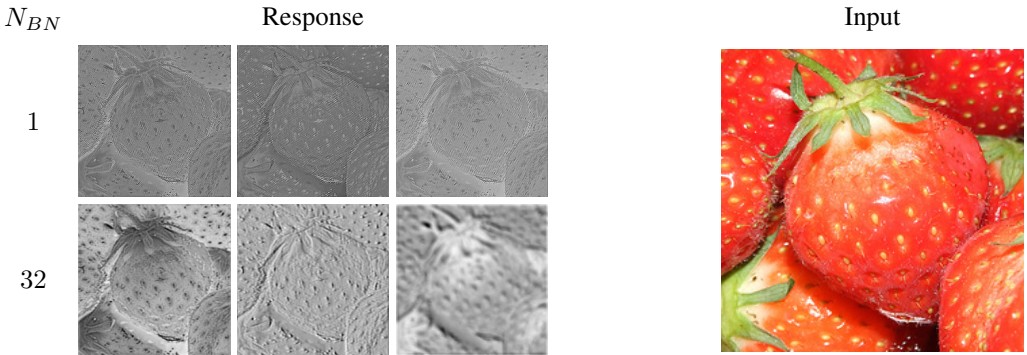

Figure 9: Cell responses to an example input for the same cells used in Figure 8 from networks with $N_{BN} = 1$ and $N_{BN} = 32$.

## A Receptive Field Visualisations

Figure 8 shows the receptive field visualisations in grey-scale and colour for cells in the Retina-2 layer of networks with $N_{BN} = 1$ and $N_{BN} = 32$. These are computed by taking the gradient of the response of the centre neuron to a blank stimulus (constant value of zero) and then applying min / max normalisation to obtain an image. The results show that the cells in networks with $N_{BN} = 1$ are somewhat centre-surround, but with quite a small extent. In contrast, the cells in networks with $N_{BN} = 32$ have much smoother receptive fields that are sensitive to changes over a much larger region of the input. Although it is true that we find exclusively centre-surround cells in the networks with $N_{BN} = 1$, we also find some centre-surround cells in the networks with $N_{BN} = 32$.

To try to better understand the functions performed by these cells, we plot the response of each cell to an example input in Figure 9. The results show that when $N_{BN} = 1$, the Retina-2 cells preserve most of the edge information in the image (with perhaps a mild edge enhancement effect) and respond in accordance with the brightness of the input (albeit inverted for two of the cells). In contrast, the Retina-2 cells in networks with $N_{BN} = 32$ distort the input heavily, performing specific colour opponent edge detection functions.

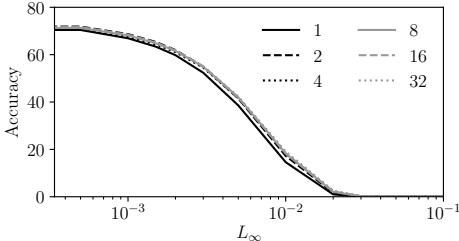

Figure 10: Robust accuracy of the CIFAR-10 trained models from Harris et al. [2019, 2020] with different bottleneck widths (line styles) against $L_\infty$ adversarial attack budgets.

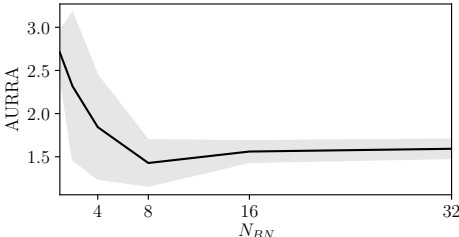

Figure 11: Model calibration, Area Under the Response Rate Accuracy curve (AURRA), on ImageNet-A [Hendrycks et al., 2019] as a function of bottleneck width. Networks with tighter bottlenecks exhibit improved calibration.

## B  CIFAR-10 Adversarial Robustness

To determine the adversarial robustness of the models trained on CIFAR-10 from Harris et al. [2019, 2020], we perform the same suite of attacks as in our ImageNet experiments (outlined in Appendix C) but over the whole CIFAR-10 test set. Figure 10 shows the worst case accuracy for these models following all attacks. The results again show that no significant difference is obtained through the introduction of a bottleneck.

## C  Artificial Adversarial Attacks

We use the following adversarial attacks with $L_\infty$ perturbation budgets between 0.0 and 1.1. Each attack uses the FoolBox [Rauber et al., 2020] implementation with default parameters.

- Fast Gradient Sign Method [Goodfellow et al., 2015]
- Projected Gradient Descent [Madry et al., 2018]
- Basic Iterative Method [Kurakin et al., 2017]
- Additive Uniform Noise
- DeepFool [Moosavi-Dezfooli et al., 2016]

## D   Model Training

Models were trained on the ImageNet 2012 training set on nodes equipped with four NVidia Quadro RTX 8000 GPUs. The following setup was used:

| | |
|---|---|
| Batch size: | 1024 |
| Optimiser: | SGD with the following: |
| initial lr: | 0.1 |
| momentum: | 0.9 |
| weight decay: | 1e-4 |
| schedule: | lr drops by factor of 10 at 30 and 60 epochs |
| epochs: | 90 |
| Data Augmentation: | Random resized cropping to 224x224 |
| | random horizontal flipping |
| | normalisation using standard ImageNet mean and s.d. |

Code and all trained models will be made available following de-anonymisation of the manuscript.

