# OpenReview forum: "Anatomically Constrained ResNets Exhibit Opponent Receptive Fields; So What?"
_NeurIPS.cc/2020/Workshop/SVRHM — SVRHM@NeurIPS Poster_

### Official Review · AnonReviewer1 · 2020-10-29
**A retina-like bottleneck does not improve robustness of learned representation in a ResNet-50 --- an interesting observation**

**Rating:** 6
**Confidence:** 4

**Review:**

This paper studies the representations learned by a ResNet-50 (trained on Imagenet) equipped with an anatomically inspired retinal bottleneck in the first layers. It appears that representations learned by this anatomically inspired network are only moderately less shape-biased than a vanilla network, and are not more robust to adversarial example attacks than a vanilla network.

It is interesting to study whether different anatomically-inspired constraints can lead to networks that are more robust than current SoTA model, and this study clearly discards the hypothesis that a retina-like dimensionality bottleneck would lead to improved representations in this model.  It is also important to report results on these investigations when they do NOT lead to better representations, so as to avoid many duplicated work.

I would thus like to recommend this paper for publication in this workshop.

Additional feedback:
- I did not understand whether this study could reproduce the results of Harris et al on color opposition, and center-surround RFs like in Lindsey et al.
- In the result section exploring superstimuli, I did not understand how it was possible to obtain stuperstimuli with more than one color axis with a bottleneck of dimension 1. How more than one color axis be conveyed in this case is unclear to me.
- The paper would be much easier to parse and the results easier to remember if the results were stated in the abstract and introduction (and consider making the the title more explicit), instead of keeping the suspense until the result section.

---

### Official Review · AnonReviewer3 · 2020-10-29
**Interesting approach and results, well written, relevant for this workshop. Potential for nice findings**

**Rating:** 7
**Confidence:** 4

**Review:**

The approach implemented in this paper has I think great potential. Given that potential, however, the results presented here (although interesting!) can seem at times slightly underwhelming and could in my opinion be improved (discussed in detail bellow). Nevertheless, the subject of the paper fits perfectly into the topic of this workshop. The paper is also well written. The authors successfully presented their results in a clear order, and discussed them in detail and with great honesty.

The work presented here follows a recent study by Lindsey et al. 2019. The referenced study showed that one could recreate some characteristics of a primate early visual system within a simple Deep Convolutional Neural Network by implementing constrains at the early stages of its processing akin to the physical and biological constrains present in the retina. More specifically, Lindsey et al. observed the emergence of a functional segregation between kernels of the last layer of the retinal-bottleneck model and kernels of the following layer. In the first case, kernels rather exhibited center-surround receptive fields, and in the second kernels exhibited receptive fields specific to specif orientations. A similar difference is found between cells in the LGN, and cells in V1.

The major novelty of the study submitted here is that they applied the same constraints to a state-of-the-art architecture (ResNet-50) much deeper and more complex that the simple architecture of the original study. They then show that when the constraint is high (1 channel bottleneck), kernels of the network show great linearity and luminance opponency (but not color). A looser constraint (>1 channels bottleneck) allows for the emergence of kernels with some degree of color opponency, both in Retina-2 and ResNet-1.
Unfortunately, the author missed at this point the opportunity to look in detail whether color opponent kernels exhibited different receptive fields in Retina-2 or ResNet-1. This would be I think a finding of great interest. The fact that early kernels of DNNs trained for object recognition on ImageNet exhibit color opponency is in itself not novel. It seems unlikely here that it is a consequence of the presence of any bottleneck. Rather, it could simply be the consequence of the input/output of these layers to be multi channeled. Works on the color representations within DNNs trained for object recognition on ImageNet have shown that deep architectures (not ResNet, but AlexNet and VGG-nets so deep nonetheless) present single and double-opponent kernels at their very early layers (see Flachot & Gegenfurtner 2018, Rafegas et al. 2018). Just looking at the first layer kernels in the original AlexNet paper (Krizhevsky et al. 2012, Figure 3), one can see that many kernels are double-opponent, with at times center-surround, at times edge-oriented receptive fields.

The higher robustness and shape bias found for networks with tight bottlenecks is quite interesting and puzzling. These findings open many questions worth investigating.

---

### Official Review · AnonReviewer2 · 2020-10-30
**Great experiment, but mixed results make me wish for more investigation.**

**Rating:** 7
**Confidence:** 3

**Review:**

This paper tests the hypothesis that deep CNNs with bottlenecks can exhibit emergent properties like luminance/color opponency and center-surround organization, as it applies to contemporaly network architectures like ResNet-50. They experiment with a bottleneck of varying size and study how it impacts emergent properties like opponency, performance, and robustness of the resulting network.

The authors report the following results. Networks with tighter bottlenecks tend to exhibit:

* Luminance opponency, but not color opponency
* Relatively higher shape bias
* Slightly lower performance as measured by top-1 accuracy
* Slight improvement in robustness to adversarial examples
* Decrease in robustness to targeted adversarial examples

These are mixed results, and while intriguing, it is hard to draw a clear conclusion. While bottlenecked networks appear to have emergent properties like color opponency and higher shape bias, it fails to translate to a definitive improvement in performance or robustness. It appears that decreased performance might just be due to a decrease in the number of parameters, and hence, a comparison against a smaller network without a bottleneck might make sense.

Yet, the work is technically sound and I support acceptance. However, I encourage the authors to further investigate the reasons behind the degradation in performance.

---

### Public Comment · ~Ethan_Harris1 · 2020-12-05
**Response to reviews**

We would like to thak the reviewers for their time and insightful comments. We have now uploaded an updated version which enhances exposition and clarity in response.

---

### Decision · Program_Chairs · 2020-11-02

Accept (Poster)